# Evaluation of Radon Action Plans: Searching for a Systematic and Standardised Method

**DOI:** 10.3390/ijerph20237128

**Published:** 2023-11-30

**Authors:** Meritxell Martell, Tanja Perko, Kateřina Navrátilová Rovenská, Ivana Fojtíková, Robbe Geysmans

**Affiliations:** 1Merience, Llimoner, 30, 08734 Olèrdola, Barcelona, Spain; 2Belgian Nuclear Research Centre SCK CEN, Boeretang 200, 2400 Mol, Belgium; tanja.perko@sckcen.be (T.P.); robbe.geysmans@sckcen.be (R.G.); 3Faculty of Social Sciences, University of Antwerp, Prinsstaat 13, 2000 Antwerp, Belgium; 4National Radiation Protection Institute (SURO), Bartoškova 1450/28, 14000 Prague, Czech Republic; katerina.rovenska@suro.cz (K.N.R.); ivana.fojtikova@suro.cz (I.F.)

**Keywords:** radon, radon action plan, indicators, review, radiation, assessment

## Abstract

Radon, a carcinogenic radioactive gas, is a leading cause of lung cancer according to the World Health Organization. European Member States are required to develop and implement National Radon Action Plans (RAPs) to address its dangerous health effects. However, assessing the effectiveness of these RAPs presents challenges for authorities. This study aims to explore the possibility of a systematic and standardised assessment method to evaluate the effectiveness of RAP strategies and its implementation. The method involved analysing the strategies of 27 EU Member States and the UK, conducting legal document analysis and group interviews with responsible authorities. Additionally, four regional workshops and one final European workshop were held. The research took place from March 2021 to May 2023. Findings indicate that evaluating RAP effectiveness is challenging due to limited existing common criteria or indicators. To address this, the study proposes guiding questions for each element required by the EU Directive, as well as additional questions related to education and training. This contribution benefits RAP owners and European regulatory authorities, supporting the development of effectiveness indicators for RAPs. By improving assessment methods, we can enhance the effectiveness of strategies in mitigating the risks associated with radon exposure.

## 1. Introduction

Radon is one of the major indoor air pollutants that comes from the decay of naturally occurring uranium and thorium in soil and rocks. It can accumulate in indoor environments through small cracks or holes in a building’s substructure. According to the World Health Organization [1], radon is one of the leading causes of lung cancer in both non-smokers and smokers, with synergistic effects in the latter. Radon is also “estimated to cause between 3% to 14% of all lung cancers in a country, depending on the national average radon level and smoking prevalence” [2]. In Europe, around 19,000 lung cancer deaths in 2019 may have been due to naturally occurring indoor residential radon [3]. Accordingly, protection against radon is one of the actions included in Europe’s Beating Cancer Plan as well as achieving a tobacco-free Europe [4].

At the international level, the *WHO Handbook on Indoor Radon* [1] highlights the need for national radon programmes, including multi-agency collaboration, the role of policymakers and authorities, financial considerations, mandatory versus voluntary approaches and establishing a national reference level. Additionally, eight organisations (European Commission, Food and Agriculture Organization of the United Nations, International Atomic Energy Agency, International Labour Organization, OECD Nuclear Energy Agency, Pan American Health Organization, United Nations Environment Programme and World Health Organization) [5] cosponsored the International Basic Safety Standards on Radiation Protection and Safety of Radiation Sources, which require International Atomic Energy Agency (IAEA) Member States to establish and implement an action plan comprising coordinated actions for controlling public exposure due to radon indoors [4]. A similar requirement is introduced at the European level with the Council Directive 2013/59/Euratom Basic Safety Standards, BSS Directive [6], which mandates that European Union Member States (EU MS) establish National Radon Action Plans (RAPs) to reduce radon exposure and ultimately the risk of lung cancer (hereinafter referred to as BSS Directive). Annex XVIII of the Directive provides a list of 14 items to be considered by authorities in preparing a RAP. RAPs should include measures such as radon mapping, the promotion of radon-resistant construction techniques to prevent radon ingress into new buildings and the provision of information to the public on radon risks and mitigation measures.

Bochicchio et al. [7] have noted the need for adequate indicators to assess the effectiveness and progress of the actions established in RAPs. They also support the use of indicators for taking decisions on future approaches to radon management. Based on our evaluation, there is currently no common set of criteria or indicators to measure the effectiveness of the different actions included in the RAPs. This paper is based on the European Commission-funded study EU-RAP entitled “Review and evaluation of national radon action plans established in EU Member States according to the requirements in Council Directive 2013/59/Euratom”. The paper sheds light on the evaluation of RAPs in EU member states and the UK. It investigates which review requirements and practices are currently in use, and provides more detailed examples from three countries that have developed and used specific assessment indicators. Furthermore, it considers whether the assessment methodology used in the EU-RAP project itself could be of value for future national RAP evaluations.

## 2. Radon Contextual Information per EU Member State and the UK

This section summarises radon contextual information in all European member states and the United Kingdom, particularly reference levels (RL), dose coefficient, the delineation of radon priority areas (RPA) and the data and criteria used for the delineation as of January 2023, as shown in Table 1.

## 3. Methods

A mixed method approach was used, comprising an online survey, content analysis of legal documents, group interviews, four regional workshops and the final workshop. The study was conducted in 27 EU MS and the United Kingdom (UK). Firstly, the online survey was sent to all responsible authorities charged with the implementation of RAPs in February 2021. The survey focused on which authorities are (or will be, as foreseen in the RAP) responsible for different radon matters and whether this responsibility is shared or not. The responses allowed the authors to identify responsible authorities for different aspects of radon management. Subsequently, reminders were sent at biweekly intervals and then weekly to authorities that had not responded to the survey. In March 2021, all authorities received a report with the information gathered in order to cross-check our understanding of their answers and to clarify any open questions. Secondly, for the legal document review and content analysis, the authors contacted national authorities in order to collect RAPs or related legal documents (if RAPs were not yet developed). A total of 25 RAPs, including draft versions of the plans, and related documents were received and analysed. A protocol for the analysis guided the extraction of data to be analysed and was consulted on with the EU-RAP reference group of experts who provided advice and expertise to the consortium and validated the project results. This group consisted of twenty experts from different disciplines involved in radon risk management. Questions were defined for each of the 14 elements of Annex XVIII of the BSS directive, and for a 15th element focused on radon-related education and training, as shown in Appendix A. For each of these 15 elements, additional relevant documents from the IAEA, WHO or other European Council Directives were consulted to formulate questions for the assessment. Coders received specific training to ensure that the same method was used when analysing the data. All aspects were coded by two independent coders. In case of different data extracted and different codes, a third (master) coder discussed the differences and decided on the final data extracted (agreed data). Thirdly, the findings from the legal document content analysis were discussed with representatives in each country through group interviews, which were conducted online due to the COVID-19 pandemic, between October 2021 and March 2022. These group interviews (between 2 and 15 people from the specific country) were conducted to validate the information as well as to respond to and clarify any missing information related to mitigation. During the interviews, one of the 14 elements in Annex XVIII that underwent assessment pertained to “reviews of the action plan”. The guiding questions were designed to address both (a) the strategy development phase for the RAP and (b) the implementation phase of the RAP, as follows:


**Strategy:**


Does the RAP define or include information regarding:Schedules/frequency of the review of the RAP?Criteria to be met on how the plan is reviewed (e.g., cost, time scales, quality, scope, benefits, risks)?Who is/are the reviewer/s of RAP?


**Implementation phase**
*:*


How does your country implement reviews in practice:When was it undertaken?Which criteria are used for the review (e.g., cost, time scales, quality, scope, benefits, risks)?How do they review results?Who was/were the reviewer/s?

Furthermore, the same guiding question, “Have you considered any indicators to measure the effectiveness of this element?”, was formulated at the end of each of the 15 elements reviewed (14 elements listed in Annex XVIII of the BSS and to be considered in the RAP strategy and education and training as an additional element considered relevant by the EU-RAP team, provided in Appendix A). Notes from the interviews were sent back to all representatives for verification and additional feedback.

Fourthly, four regional workshops were held to examine similarities and differences in the approach followed to evaluate RAPs and the type of indicators used. The main focal points at the workshops included: (a) Would it be feasible to have a set of common criteria for all EU MS to review the RAP or should these criteria be country specific? (b) Reflection on the advantages and disadvantages of the owner as the reviewer. The four workshops were conducted between October 2021 and March 2022. Finally, the overall results were discussed, verified and compared at a final workshop in Brussels, Belgium, with 50 participants from various EU countries and the United Kingdom in September 2022. The workshop covered topics such as ‘*Who, when and what to review*?’ and ‘*Is a set of common indicators for European MS and the UK appropriate*?’.

## 4. Results

As of May 2023, all EU MS and the UK, except Italy and Spain, had approved their RAPs. A few countries in Europe have already a set of criteria or indicators which facilitate the review of their national radon action plans, e.g., Belgium, France, Ireland, and Portugal. The review of RAPs is mostly based on the degree of completion of the actions planned in the RAPs. Few countries have defined indicators helping assessment of their RAP’s effectiveness, which may include:-Number of measurements conducted in dwellings and workplaces annually (e.g., Belgium);-Number of awareness-raising activities (e.g., Slovenia);-Share of remediated buildings reported (e.g., Slovenia) or annual number of remedial actions reported (e.g., Belgium);-Assessment of the cost-effectiveness of the interventions (e.g., Ireland) or effectiveness of remedial actions reported (e.g., Belgium);-Findings of the annual inspections programme (e.g., Belgium);-Number of courses held and attendance at radon prevention training (e.g., Ireland).

Several countries are working to define effectiveness indicators in the short and long term (e.g., Austria, Germany). One of the measures included in the RAP in Germany includes the need to develop and identify short and long-term indicators to evaluate radon protection measures. The plan states that “short and long-term review criteria to evaluate the success of the measures and targets for protection against radon in Germany envisaged under the RAP and radiation protection legislation shall be set out. Review criteria are to be developed in the form of measurable indicators. These indicators shall facilitate a review of target achievement of measures before an update of the radon action plan” foreseen approximately in 2028. 

However, there is not yet a common set of criteria or indicators to assess the effectiveness of the different elements of the RAP. Bochicchio et al. [7] summarise the activities undertaken by Heads of European Radiological Protection Competent Authorities (HERCA) on effectiveness indicators. The HERCA working group on Natural Radiation Sources decided at the second workshop on national RAPs held in June 2022 that “a list of indicators is useful, but selection and applications depend on national circumstances; the need for description of which input data are required was identified. HERCA will not endorse a harmonised list of indicators that must be used as these may be quite different depending on countries’ prevailing circumstances, available resources, etc. HERCA representatives concluded that, at the moment, no need for a harmonised European list is foreseen. Instead, the exchange about indicators will be continued in coming meetings” [8], p. 5.

During the four EU-RAP regional workshops, the interest in establishing a set of indicators to evaluate the RAPs was raised. The need to recognise the differences in the context of RAPs and incorporate country specificities was also highlighted. For some participants, the guiding questions used in the EU-RAP project to assess the implementation phase for each of the 14 items defined in Annex XVIII of the Council Directive 2013/59/Euratom and education and training proved to be a useful mechanism to help countries evaluate the effectiveness of the actions included in the corresponding RAP.

The schedule to review RAPs varies widely among countries and ranges from 2 to 10 years. In some cases, RAPs do not specify a review schedule, and the interviewees indicated that the reviews are conducted when deemed necessary. This raises the question of what criteria define this need and who is responsible for signalling it. Typically, the responsible organisation for RAP development also serves as the reviewer. When this is not the case, there may be undefined reviewers, multiple organisations involved in the RAP might share or decide on the reviewing responsibility or independent experts may be appointed as reviewers (e.g., Greece).

Table 2 summarises who is the reviewer of the RAP, the schedule and the criteria for the review for each country. The specific indicators used in France, Ireland and Portugal are described in detail below.

In the following section, current assessment practices and indicators in various European countries are presented in more detail. Focus is put on those countries which have defined explicit indicators for evaluating their RAPs.

### 4.1. Indicators in France

In France, a system of specific indicators has been put into place to evaluate the effectiveness of the national strategy implemented under the RAP [9] which aims to evaluate the effectiveness of the RAP on a yearly basis and in the long-term. The indicators, shown in Table 3, were chosen for their pertinence and the available data enabling them to be monitored. Measuring the health impact via the change in the number of radon-induced lung cancers can only be evaluated over the long term. Similarly, data on the average indoor radon concentration in dwellings, workplaces and buildings open to the public, reflecting the exposure of the population, are only available on a long-term basis. This requires the determination of intermediate indicators allowing indirect evaluation of the reduction in exposure. At present, the aim is to monitor the implementation of the regulations per sector: general public, workplaces and buildings open to the public. These indicators will ultimately be monitored over the long term.

The review of the RAP is foreseen every 4 years, and the first assessment will be published in 2025. The assessment is published online. A pilot committee assesses each action and shares the results with the steering committee.

### 4.2. Indicators in Ireland

The review of the RAP was undertaken on a yearly basis in the past and is now every 2.5 years. The reviewer is a coordination group led by the Environmental Protection Agency (EPA Ireland). There are two types of indicators used [11]:-Leading indicators: These give a real-time measure of progress towards reducing exposure. These indicators can be used as reliable evidence that the long-term objective will be achieved. These indicators include the number of domestic radon tests; number of radon tests linked to conveyancing; remediation rate; rate of successful outcomes for those who remediate; number of courses held and attendance at remediation training; number of businesses that include radon in Health and Safety assessment and website hits.-Lagging indicators: These complement the leading indicators and provide information that may not be sufficiently timely to helpfully direct ongoing actions. These indicators include population-weighted national average indoor radon concentration; geographically weighted national average indoor radon concentration and radon awareness levels.

The criteria for the review are the status of each action; the impact of the measures taken to date; the likely effectiveness of the strategy in the longer term in reducing the risk from radon; an updated assessment of the cost-effectiveness of the interventions recommended in the strategy; review of stakeholder experience of the strategy; lessons learned and outstanding issues; and identification of further actions appropriate at that time.

### 4.3. Indicators in Portugal

In Portugal, the Environmental Protection Agency (APA) will review the RAP every 5 years. To support evaluation, a set of metrics consisting of two types of indicators are considered: (1) core or efficiency indicators, which refer to the achievement of the measures within the stipulated timeframe; (2) secondary or effectiveness indicators (these indicators are complementary to the core indicators that provide evidence that the long-term objective will be achieved). A private company linked to the university was appointed to help develop the matrix of indicators to evaluate the RAP. This matrix was then reviewed by several institutions.

The criteria and secondary indicators in the RAP in Portugal [12] relate to the following three dimensions:(a)Radon exposure of the population: it assesses the contribution of the RAP in reducing the occurrence of adverse effects on human health from long-term exposure to radon (Table 4)-Health risks—assesses the risks to which the population is exposed through an epidemiological study.-Workers’ exposure—assesses the mechanisms for managing radon in workplaces and the protection of workers.-Demographic structure of the population exposed to radon—assesses the age structure of the population, the class, gender and geographical distribution of the population exposed to radon.(b)Quality of the building stock: it assesses the contribution of the RAP in improving the characteristics of the building stock (housing and workplaces) for radon protection, both in the construction of new buildings (preventive measures) and in existing buildings (remedial measures) (Table 5).-Buildings—assesses the distribution of buildings, their age of construction, and the existence of construction features (heating, isolation).-Constructive solutions—evaluate existing regulations and standards in relation to the constructive guidelines.-Housing stock costs—evaluates the costs of housing stock real estate with preventive measures in place.-Energy efficiency—evaluates the relationship between saving measures emissions that are in place and indoor air quality.(c)Governance: it assesses the level of articulation and capacity development of the entities involved in radon management (Table 6).-Institutional articulation—evaluates how the existing institutional articulation allows the management of radon, defines responsibilities, defines competencies in the management of ionising radiation and if there are financial resources for the implementation of the plan.-Technical skills—assesses the existing mechanisms for the technical capacity of actors involved in radon management, namely professionals from public institutions.-Accreditation of measurement and mitigation services—assesses the levels of standardisation/accreditation of existing service providers and certification of building materials.-Awareness raising among society and stakeholders—evaluates how the plan contributes to disseminating information and raising awareness among the population and stakeholders.

While the list of countries which have developed specific indicators is currently not extensive, it might serve as inspiration for other countries to further develop their RAP reviews. Additional inspiration might also be found in the EU-RAP project itself, which has, as described above, conducted an extensive review of current RAP development and implementation across EU member states and the UK.

### 4.4. The EU-RAP Methodology

The EU-RAP study utilised and examined a list of guiding questions as part of a systematic and standardised assessment approach to evaluate the efficacy of RAP strategies and their implementation. The guiding questions for each of the 15 assessed elements are listed in Appendix A.

## 5. Discussion

This study presents guiding questions aimed at assessing the strategy and its implementation for each element outlined in national RAPs, as mandated by the EU Directive. Additionally, it includes supplementary questions pertaining to education and training aspects, as shown in Appendix A.

The study findings reveal that while approximately half of the EU Member States have established a set of indicators in their RAPs to assess the effectiveness of the included actions, only a few countries have conducted evaluations of their RAPs. Furthermore, these evaluations were limited to specific elements, lacking clear indicators for each element. We also observed a lack of indicators connecting indoor air quality and national programmes focused on reducing energy consumption in buildings, conducting thermal retrofitting and maintaining indoor air quality in both dwellings and workplaces. Similar to the case of smoking cessation programmes, the absence of robust connections between RAPs and initiatives for air quality and energy efficiency programmes could represent a missed opportunity.

Completeness of the actions is the most common criterion to review the RAPs. Annual follow-ups of the progress of the tasks in the RAPs (in addition to reviews) are foreseen in most countries. Since concluding this study at the end of 2022, countries have updated their RAPs, while the data remain valid.

This paper has presented an overview of the review characteristics, including the reviewer, schedule and criteria for all 27 EU MS and the UK and has provided the detailed indicators included in the RAPs of France, Ireland and Portugal. The article proposes a set of questions to evaluate the effectiveness of RAP strategies and their implementation. When asked in the regional workshops whether it would be feasible to have a set of common criteria for all EU MS, most countries agreed that it would be useful to set some common goals and would urge countries to critically evaluate their RAPs. Nevertheless, any attempt to homogenise indicators is challenging—and sometimes undesirable–due to the different stages of implementation of RAPs, the contextual, cultural and geographic differences and the specific actions set in the different RAPs. Overall, two types of criteria could be used to assess the effectiveness of RAPs: a common set of criteria for all EU MS and country-specific criteria.

In this study, we observed a lack of indicators related to smoking cessation and the absence of connections between radon exposure and anti-smoking campaigns in the prevention of radon-induced lung cancers. This lack of indicators and connections could be a missed opportunity to integrate radon exposure information with anti-smoking initiatives to prevent lung cancers.

The purpose of the evaluation is to make a judgement about the level of implementation of RAPs, to improve their effectiveness and/or to inform programming decisions. The fact that in most countries the owner is the reviewer of the RAP has advantages since it is the organisation which has the information and knows better about the progress made and the actions achieved. Nevertheless, external independent evaluators and/or peer reviews could definitely aid and complement the evaluation.

A common set of indicators to evaluate the effectiveness of RAPs would be of benefit to the owners of RAPs and to cross-national organisations, like HERCA. Obviously, the country context (e.g., climate, geology, building practices, economic situation, prevailing country-specific radon risks, etc.) should be taken into account when using and comparing these indicators. A set of guiding questions for the implementation phase as the one proposed in the EU-RAP study proved to be a useful mechanism to help countries evaluate the effectiveness of the actions included in the RAP. Ultimately, RAPs should serve to improve public health by reducing radon-induced lung cancer and in this regard, indicators showing the level of smoking decline and the uptake of radon testing and mitigation are key to evaluating the effectiveness of RAPs.

## 6. Conclusions

Overall, the research highlights the different practices undertaken by authorities in European countries to review RAPs. Our results suggest that a common understanding of the scope and the purpose of the reviews of RAPs could be useful for EU MS. Whilst the evaluation of the RAPs is nationally based, a European methodology such as the one developed by EU-RAP which can be adapted to the national circumstances would be particularly important and would urge authorities to take a common approach to assess effectiveness. Sharing information on indicators and ways to evaluate RAPs is extremely beneficial for national authorities as well as cross-national organisations such as HERCA.

Lessons learned from this analysis include:-Indicators for assessing the effectiveness of radon actions may refer to the level of achievement of the measure within a stipulated timeframe or to the long-term objective.-Many countries currently lack specific indicators to assess their RAP, and a common set of indicators could be helpful in this regard. This common set should, however, be developed and used taking into account contextual differences.-The utilisation of a set of questions to assess the effectiveness of RAP strategies and their implementation has proven to be a valuable tool.

## Figures and Tables

**Table 1 ijerph-20-07128-t001:** Reference levels, dose coefficient, approach, data and criteria used for the delineation of RPA in EU MS and the UK.

Country	Reference Level	Dose Coefficient	RPA Defined	Data Used for Delineation of RPA	Criteria for Delineation
Austria	300 Bq/m^3^	ICRP137	Yes	Measured indoor levels combined with certain user behaviour/building characteristics	Models the mean predicted radon concentration in a municipality
Belgium	300 Bq/m^3^	ICRP137	Yes	Indoor measurements in dwellings	Municipalities where the probability of exceeding the RL of 300 Bq/m³ is above 5%
Bulgaria	300 Bq/m^3^	ICRP137	In progress	Indoor radon levels	Exceedance of RL
Croatia	300 Bq/m^3^	/	Not finished	Radon concentrations in soil	10% of buildings above RL
Cyprus	300 Bq/m^3^	ICRP137	Not planned due to prevailing geological situation		
Czech Republic	300 Bq/m^3^	ICRP137	Yes	State of radon concentration indoors and in the soil, geological parameters of subsoil, age of building stock	30% of buildings above RL
Denmark	100 Bq/m^3^ (except new build)	ICRP137	Yes	Specific indicators for situations with potentially high radon exposure are of a constructional or geological nature, including age of construction, type of building or local soil conditions	
Estonia	300 Bq/m^3^ (pre)schools: 200 Bq/m^3^	ICRP137	Yes	Radon concentration in approx. 13,000 dwellings and a methodology that incorporates geological and environmental dose rate information	Over 10% of dwellings on the ground floor above 300 Bq/m^3^. For new dwellings: list of municipalities drawn up into three groups (no risk, medium risk and potentially high exposure risk)
Finland	300 Bq/m^3^ New buildings: 200 Bq/m^3^	ICRP137	No	Finland’s residential areas are not divided into different radon risk areas since on the whole territory high radon levels can be measured	
France	300 Bq/m^3^	New order ICRP137	Yes	Capacity of the soil to emit radon, geological maps	Type of bedrock and cracks in the soil
Germany	300 Bq/m^3^	ICRP65 but 137 under discussion	Yes	Local distribution of radon activity concentration in the soil gas, soil gas permeability, radon activity concentration indoors, local data of individual federal states such as soil and rock type	Criteria laid out in the legislation: RPA where 10% of the buildings in at least 75% of the given administrative area are defined as radon risk
Greece	300 Bq/m^3^	ICRP137	In progress	Radon activity concentration collected through national radon survey and Ra-226 measurements in water and soil	‘Priority area’: geographical area in which the probability that a radon concentration exceeding the RL will be found in a single-store dwelling > 10%
Hungary	300 Bq/m^3^	ICRP137	No, some areas already known	Indoor radon from representative radon survey and geology	Will be selected based on mapping results
Ireland	Workplaces: 300 Bq/m^3^ Schools, dwellings (advised): 200 Bq/m^3^	ICRP137 under discussion	Yes	32,000 indoor radon measurements (geocoded) with variables including bedrock geology, soil geology, soil permeability and aquifer type	Logistic regression model, 10% of dwellings predicted to be above the RL
Italy	300 Bq/m^3^	ICRP137	In progress	Indoor radon data in dwellings, workplaces and schools; geological map of Rn risk indoor is being developed to help regions that do not have any data to start delineation	Level of 300 Bq/m3 is exceeded in 15% or more dwellings on the ground floor
Latvia	Dwellings: 200 Bq/m^3^ 300 Bq/m^3^ Workplaces: 400 Bq/m^3^	/	No	The national RAP includes the following criteria for delineation: (1) geological situation of the administrative territory; (2) during the radon level assessment, at least in 80% of buildings in the administrative territory the indoor radon gas level exceeds 200 Bq/m^3^.	
Lithuania	300 Bq/m^3^	ICRP137	Yes	Indoor measurements	Area where the average annual Rn concentration indoors exceeds 300 Bq/^3^ in 10 % of all buildings measured
Luxembourg	300 Bq/m^3^	ICRP137	Yes	Indoor measurement results from more than 5000 buildings used	Fraction of houses exceeding the RL indoor radon value (5% above RL)
Malta	300 Bq/m^3^	No need	Not planned due to prevailing geological situation		
Netherlands	100 Bq/m^3^	ICRP65 but ICRP137 under discussion	Not planned due to prevailing geological situation		
Poland	300 Bq/m^3^	Decision to be made	Yes	Geological structure, content of Ra-226 in the ground, tectonic structure, erosion zones	Uranium concentration in the structures is found above 4 g/t (4 ppm), and where radon concentration in water is above 100 Bq/l
Portugal	300 Bq/m^3^	ICRP65 but ICRP137 under discussion	Yes	Geogenic variables	Low-risk areas—RL not expected to be exceeded in > 10% of buildings; Moderate-risk areas—RL can be exceeded in > 10% of buildings; High-risk area—RL is exceeded in > 10% of buildings;
Romania	300 Bq/m^3^	ICRP137	Not yet	Indoor radon; method from JRC radon atlas	Insignificant risk <5% of buildings above RL; Medium risk >5% and <10% of buildings above RL; High risk >10% of buildings above RL
Slovakia	300 Bq/m^3^	ICRP137	Not yet	Indoor radon concentration from representative radon survey	Will be selected based on mapping results
Slovenia	300 Bq/m^3^	ICRP137	Yes	Measurements of indoor and in-soil gas, geological structure, historical measurements, geological compositions, Ra-226 content, soil permeability	30% of buildings above RL
Spain	300 Bq/m^3^	ICRP137	Yes	Radon concentration in approx. 13,000 dwellings and geological and environmental dose rate information	Over 10% of dwellings on the ground floor above 300 Bq/m^3^. Three groups: no risk, medium risk and potentially high exposure risk
Sweden	200 Bq/m^3^	ICRP137 under discussion	Yes	Uranium content in the upper surface of the ground made by Geological Survey of Sweden (SGU), radon in soil concentration, soil permeability, soil type	
UK	Workplaces: 300 Bq/m^3^ Dwellings: 200 Bq/m^3^	ICRP65	Yes	Indoor measurement and geological features collected in databases	At least 1% of homes are expected to be above 200 Bq/m^3^

**Table 2 ijerph-20-07128-t002:** Review of RAPs: reviewer, schedule and criteria in EU MS and the UK (status in January 2023).

Country	Reviewer of RAP	Schedule for Review of RAP	Criteria for Review of RAP
Austria	Federal Ministry of Climate Action, Environment, Energy, Mobility, Innovation and Technology (in discussion with competent authorities)	Every 10 years and in the event of substantial changes. Review foreseen in 2031.	To be developed to identify any changes in the state of knowledge.
Belgium	Federal Agency for Nuclear Control, FANC	Every 5 years	Annual number of measurements performed in dwellings and workplaces. Evolution of the statistics of the performed measurements. Annual number of remedial actions reported. Effectiveness of the remedial actions reported. Findings of the annual inspection programme.
Bulgaria	National Coordination Council	5 years (foreseen end of 2022)	Performance evaluation and cost-effectiveness analysis. Results/indicators defined for every action
Croatia	Intention to involve international experts. Feedback from EU-RAP and IAEA Technical Cooperation project would be useful	Every 5 years, but undertaken in 2022	/
Cyprus	Not defined	When necessary (potentially 2025)	/
Czech Republic	The State Office for Nuclear Safety, SÚJB	Every 5 years To be reviewed in 2024–2025	/
Denmark	Housing and Planning Agency initiated the review. Monitoring group consisting of representatives from different building owners, building industry partners, universities and research organisations and authorities	RAP reviewed when completed	Completion and efficiency of the actions
Estonia	Ministry of Environment	Every 2 years	Completion of activities planned
Finland	Steering Committee	Every 5 years	Annex I of RAP defines recommendations and responsible parties (not indicators)
France	Oversight Committee	Every 5 years and on a yearly basis	General indicators and indicators for each action to evaluate the effectiveness of the RAP.
Germany	BMUV, with consultation with federal states	At least every 10 years	A system of indicators being developed through a research project.
Greece	Independent group (not yet set)	Every 10 years (interim 5th-year evaluation); 3 years revision	Effectiveness of the whole procedure is part of the deliverables and timescale.
Hungary	Ministry for Human Capacities, Ministry for Innovation and Technology and Prime Minister’s Office	Every 4 years To be reviewed in 2023	Objectives met; methodologies appropriate; new scientific and technical methods introduced; regulatory framework appropriate
Ireland	Department of Environment, Climate and Communications in collaboration with EPA and the National Radon Control Strategy Coordination Group	Every 5 years	At the end of the 5 years, the coordination group will carry out a detailed review.
Italy	National Radon Observatory	At least every 10 years (period reports every 2 years)	All actions have indicators to evaluate effectiveness
Latvia	Ministry of Environmental Protection and Regional Development	In 2030–2031 or when necessary based on the national radon action plan	Large-scale measurements of radon concentration in dwellings, workplaces and public buildings will be conducted in 2030–2031, when it will be assessed whether a RAP needs to be developed.
Lithuania	Radiation Protection Centre RSC	Every 5 years	All actions have indicators
Luxembourg	Division of Radiation Protection	Not fixed, but likely 2024–2025	To be developed, most likely level of accomplishment of actions
Malta	Radon Working Party (RWP)	Schedule for review of RAP should be defined not later than 2023. Review in 2022.	/
Poland	Ministry competent for health in cooperation with the Chief Sanitary Inspectorate. The Panel shall periodically assess the RAP.	Every 4 years	Completeness (whether the plan covers the required areas) and validity (whether the plan requires any amendments or updates).
Portugal	Portuguese Environment Agency (APA)	Every 5 years	Matrix of core/efficiency and secondary/effectiveness indicators
Romania	Interministerial committee	Every 5 years (in 2023)	/
Slovakia	Public Health Office (annual meeting with stakeholders to share information on set goals and tasks) (owner)	Evaluation to be submitted in 2024 Update for 2027–2031	Reduction in the number or % of dwellings with radon volume activity above the RL or reduction in the average volume activity of radon in residential areas and workplaces
Slovenia	Slovenian Radiation Protection Administration	Every 5 years (next in 2024–205)	Nº of measurements of radon concentrations per year and nº of measurements looking for radon source; nº of activities related to public awareness of radon; share of successfully rehabilitated buildings
Spain	Ministry of Health in cooperation with implementation committee	Every 5 years (annual monitoring reports)	Three-fold evaluation: process, outcome and structure. Indicators of execution and performance indicators.
Sweden	Led by SSM. Radon Group reports on status of ongoing measures and plans to steering group.	Every 5 years	Defined based on actions proposed. Yearly timetable by the radon group.
The Netherlands	To be determined	Every 10 years	To be developed
United Kingdom	Initiated by Department of Health and Social Care (funding organisation) and supported by UKHSA	Within 5 years of publication	To be developed

**Table 3 ijerph-20-07128-t003:** Indicators in the RAP in France.

1	Public buildings	1a —Number of buildings open to the public screened, as part of the regulatory monitoring of public exposure (according to the Public Health Code). 1b —Number of buildings exceeding the reference level of 300Bq/m^3^. 1c —Number of buildings exceeding the threshold of 1000 Bq/m^3^. 1d —Number of buildings in which concentration reduction work has been carried out (the effectiveness of the reduction work has been measured). 1e —Number of buildings in which additional measurements have been carried out as part of building expertise (identification of radon sources, pathways and transfer routes/paragraph 6.2 of ISO 11665-8: 2019 [10]).
2	Workplaces	2a —Number of workplaces with a result exceeding the reference level of 300 Bq/m^3^ after concentration reduction work. 2b —Number of workers who benefit from radon exposure individual dosimetry monitoring. 2c —Number of workers who have exceeded the effective dose of 20 mSv effective dose over 12 consecutive months. 2d —Number of radiation protection advisors trained on radon.
3	General public	3a —Number of local information interventions on radon which follow the design of the Ministry of Health. 3b —Number of dwellings screened during local radon information operations. 3c —Perception of radon risk among the French population.

Source: Adapted from ASN [9].

**Table 4 ijerph-20-07128-t004:** Criteria and indicators related to radon exposure of the population in Portugal.

Criteria	Indicators
Health risks	(1) % of reduction in annual radon concentration (2) Number of prevalence of lung cancer/neoplasms due to radon
Workers’ exposure	(3) Number of workers’ exposure to doses exceeding 6mSv (4) Number of protective measures (5) Number of workplaces tested (6) Number of remediated workplaces
Demographic structure of the population exposed to radon	(7) Number and % of age structure of the population (8) Number and % of distribution of the population by sex (9) Number and % of distribution of population by susceptible areas

Source: Adapted from APA [12].

**Table 5 ijerph-20-07128-t005:** Criteria and indicators related to the quality of the building stock in Portugal.

Criteria	Indicators
Buildings	(10) Number of buildings by age of construction and materials used in construction (11) State of conservation of buildings
Constructive solutions	(12) Number of building regulations and standards
Housing stock costs	(13) Construction price m^2^ (€)
Energy efficiency	(14) Number and % of buildings with energy-saving measures and radon concentrations higher than 300 Bq/m^3^

Source: Adapted from APA [12].

**Table 6 ijerph-20-07128-t006:** Criteria and indicators related to governance in Portugal.

Criteria	Indicators
Institutional articulation	(15) Organisational structure (16) Number of human resources (17) Financial Allocation/Costs (€) (18) Financial support for testing and mitigation (€)
Technical skills	(19) Number of technical documents supporting radon management (20) Number of training/awareness-raising actions for main actors (21) Number of specialists in radon management (22) Number of radon mitigation specialists
Accreditation of measurement and mitigation services	(23) Number of accredited/recognised services (24) Number of accredited/recognised companies (25) Number of certified materials
Awareness raising among society and stakeholders	(26) Number of communications in the media (27) Number of communications directed to target audiences (28) Number of engagement actions for target audiences (29) Number of stakeholder associations for radon

Source: Adapted from APA [12].

## Data Availability

The data that support the findings of this study are available from the corresponding author upon request.

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
