# Peer review of "Evaluation of Radon Action Plans: Searching for a Systematic and Standardised Method"

_ijerph, 2023, doi:10.3390/ijerph20237128_

Round 1
Reviewer 1 Report
Comments and Suggestions for Authors
This study analyses the effectiveness of Radon Action Plans rolled out by the national authorities of 27 EU Member States and the UK to assess a systematic and standardised assessment method to evaluate the strategies and their implementations.
General comments
The validity of the diagnosis has probably varied over time.
The conclusions of this analysis are not surprising such as the lack of common specific indicators among EU Member States and the UK to assess the RAPs because the basis of these strategies are focused on radon mapping, the promotion of radon resistant construction techniques to prevent radon ingress into new buildings. In my opinion, the most important indicator that should be more taken into account is not highlighted : smoking prevention.
"radon is one of the leading causes of lung cancer" Radon is the second cause of lung cancer after smoking but only 0,9% of people died from radon-induced lung cancer and more than 85% are smokers. The problem is that at the moment the case-control studies in the general population have not proved informative on this issue and valid signatures at the histologic and molecular levels have not yet been identified. In the absence of definite diagnosis the optimal strategy for reducing radon-induced lung cancer in terms of public health should focus on smoking prevention and cessation. In the US, studies showed that smoking cessation actually dominated the remediation of high-radon homes : smoking decline would alone reduce the risk of radon by half.
Conclusion
This study should highlight the necessity to focus RAPs strategy on smoking prevention and cessation first. It would be more effective and efficient in terms of public health and economic efficiency.
Specific comment
- Line 22 : correct "BY" as "By"
Reviewer 2 Report
Comments and Suggestions for Authors
Manuscript ID: IJERPH-2650623
Full Title: Assessing the effectiveness of radon action plans: Searching for a systematic and standardized method
General Comments:
The authors of the current paper set out to describe National Radon Action Plans put forth across the EU and the UK in 2021, and to determine whether there are common criteria for assessing the efficacy of these action plans from one country to another. Although such work is important and interesting, I don’t think that this manuscript is appropriately detailed or organized to address the authors’ purported goals. I do wonder from the presentation of information whether a brief review format would be a better format for the work rather than posing this as an empirical study and trying to meld the information into that format. I do also think that the authors need to consider providing an appendix with the questions and criteria they assessed rather than putting everything into the main text. As it is, roughly one-third of the total document provided is a bullet-point list of questions rather than actual useful content and description of the acquired data. I struggled to decipher any meaningful findings from the information presented in this paper. I strongly recommend that the authors reformat their work and focus on the actual data. It would be good to provide some actual analyses as well – it seemed like there was limited actual analysis of the seemingly massive amount of data acquired.
Reviewer 3 Report
Comments and Suggestions for Authors
My suggestion is to change the title of the manuscript to be "Evaluation of '''''''"
Author Response
|
Thank you very much for taking the time to review this manuscript. Please find the detailed responses below and the corresponding revisions/corrections highlighted/in track changes in the re-submitted files.
Point-by-point response to Comments and Suggestions for Authors |
|
Comment: My suggestion is to change the title of the manuscript to be "Evaluation of '''''''" |
|
|
|
Response: Thank you for your suggestion. We agree with this comment. Therefore, we have change the title to “evaluation of radon action plans: searching for a systematic and standardized method”.
|
Reviewer 4 Report
Comments and Suggestions for Authors
I read and reviewed the paper entitled “Assessing the effectiveness of Radon Action Plans: searching for a systematic and standardized method?” by Martell et al. This is a study investigating the possibility for a systemic and standardized assessment method to evaluate the effectiveness of RAPs strategies and its implements of 27 EU member States and UK. The aim of the manuscript is well described. However, there is still room to be updated as I have found major and minor issues that should be checked before publication.
- The method, results and discussion should be rewritten. It is very difficult to understand in the present manuscript.
- The author should provide and discuss more about the radon concentration activity including radon mapping and reference levels of 27 EU member states and UK.
- lines 138, 143: Please adjust.
- I suggest minor English revision because there are some expressions and sentences with wrong syntax or grammar.
Comments on the Quality of English LanguageModerate editing of English language required.
Round 2
Reviewer 1 Report
Comments and Suggestions for Authors
All comments were taken into account by the authors. Answers are satisfactory. The paper can be published.
Reviewer 2 Report
Comments and Suggestions for Authors
The authors have made adequate changes to the paper, and I thank them for taking the time to address my concerns regarding formatting and presentation.
Reviewer 4 Report
Comments and Suggestions for Authors
The authors substantially revised the manuscript according to suggestions.